# Differential Expression of the Sphingolipid Pathway Is Associated with Sensitivity to the PP2A Activator FTY720 in Colorectal Cancer Cell Lines

**DOI:** 10.3390/jcm10214999

**Published:** 2021-10-27

**Authors:** Peter Sciberras, Laura Grech, Godfrey Grech

**Affiliations:** Department of Pathology, Faculty of Medicine and Surgery, University of Malta, MSD 2080 Msida, Malta; peter.sciberras.16@um.edu.mt (P.S.); laura.grech@um.edu.mt (L.G.)

**Keywords:** PP2A, colorectal cancer, sphingolipid pathway, predictive, biomarkers

## Abstract

Protein phosphatase 2A (PP2A) is a ubiquitously expressed intracellular serine/threonine phosphatase. Deregulation of PP2A is a common event associated with adenocarcinomas of the colon and rectum. We have previously shown that breast cancer cell lines are sensitive to the PP2A activator FTY720, and that sensitivity is predicted by high Aurora kinase A (AURKA) mRNA expression. In this study, we hypothesized that high relative AURKA expression could predict sensitivity to FTY720-induced apoptosis in colorectal cancer (CRC). The CRC cell lines NCI H716, COLO320DM, DLD-1, SW480, and HT-29 show a high relative AURKA expression as compared to LS411N, T84, HCT116, SW48, and LOVO. Following viability assays, LS411N, T84, HCT116, and SW480 were shown to be sensitive to FTY720, whereas DLD-1 and HT-29 were non-sensitive. Hence, AURKA mRNA expression does not predict sensitivity to FTY720 in CRC cell lines. Differentially expressed genes (DEGs) were obtained by comparing the sensitive CRC cell lines (LS411N and HCT116) against the non-sensitive (HT-29 and DLD-1). We found that 253 genes were significantly altered in expression, and upregulation of CERS4, PPP2R2C, GNAZ, PRKCG, BCL2, MAPK12, and MAPK11 suggests the involvement of the sphingolipid signaling pathway, known to be activated by phosphorylated-FTY720. In conclusion, although AURKA expression did not predict sensitivity to FTY720, it is evident that specific CRC cell lines are sensitive to 5 µM FTY720, potentially because of the differential expression of genes involved in the sphingolipid pathway.

## 1. Introduction

With more than 1.9 million newly diagnosed cases in 2020 and nearly 935,000 deaths worldwide, colorectal cancer (CRC) is the third most prevalent cancer in men, and the second most commonly occurring malignancy in women [1]. Despite considerable advances in early detection and treatment, CRC is one of the cancers whose incidence is increasing globally, comprising 11% of all new cancer diagnoses [2]. Effective preventive strategies [3], early detection [4], and molecular typing guiding personalised treatment [5] are necessary to reduce the impact of this disease. A multilevel approach is necessary to prevent the development of CRC, and detect the presence of the condition in its early stages, including the implementation of policies and lifestyle recommendations [6].

CRC is a heterogeneous disease with significant inter-patient differences in therapy response in terms of efficacy and toxicity, partly owing to molecular diversity [7,8]. Systemic chemotherapy in the adjuvant and advanced settings has evolved considerably in the past decades from basic cytotoxic agents to combination regimens and, more recently, the introduction of biological agents targeting specific signaling pathways [9,10]. However, notwithstanding all the evidence and current recommendations, drug resistance remains a widely unresolved issue [11], as only 10% of patients benefit from adjuvant chemotherapy.

To avoid unnecessary side effects and healthcare costs, oncological treatments should be reserved only for those patients who would benefit [12]. However, the predictive biomarkers available to aid clinicians in making treatment decisions in CRC are limited. Further research is therefore needed to address the urgent need to provide personalised treatment in CRC. A consensus molecular classification (CMS) of CRC was recently adapted for preclinical models to identify subtype-specific drug sensitivities [13]. Establishing a robust molecular signature that can be applied in the clinic will allow clinicians to tackle clinical dilemmas in treating CRC patients in order to successfully implement point-of-care genomics, and provide personalized care [14]. Implementing non-invasive techniques to predict early metastatic disease requires these molecular signatures to predict disease development and progression [15]. These markers include proteins in plasma such as resistin [16], mRNA in blood-derived cells or extracellular vesicles [16], and methylation levels in faecal samples or blood [17].

Protein phosphatase 2A (PP2A) is a ubiquitously expressed intracellular serine/threonine phosphatase that maintains physiological cell function by counteracting kinase-mediated phosphorylation. PP2A plays an important role in tumor suppression [18], and decreased activity of PP2A is observed in many types of malignancies, including CRC and breast cancer [19]. Research has shown that PP2A is frequently inactivated in CRC patients [20], suggesting that PP2A represents a potential therapeutic target in CRC. Reactivation of PP2A by agents such as FTY720 can result in inhibition of cell proliferation, cell cycle arrest, and apoptosis of cancer cells [21].

FTY720 (Fingolimod, Gilenya^®^), a structural analogue of sphingosine developed from the fungal metabolite myriocin [22], is an immunomodulator mostly used in multiple sclerosis and multiple organ transplantation [23]. The immunosuppressive activity of FTY720 has been suggested to be related to its phosphorylation by sphingosine kinase 2 (SphK2), which results in the modulation of G-protein coupled sphingosine-1-phosphate receptors (S1PRs). This gives rise to lymphopenia by altering lymphocyte trafficking [24]. Apart from its immunosuppressive effects, FTY720 also demonstrates anti-cancer properties. Various in vitro and in vivo studies showed that FTY720 induces growth arrest and apoptosis in multiple types of cancer, including CRC [25]. 

The mechanisms responsible for FTY720-induced cancer cell death are poorly understood, and appear to vary according to the type of cancer. FTY720 was found to moderately inhibit the survival of CRC cell lines with a half-maximal inhibitory concentration (IC_50_) of 5 mmol/L. It was also shown that FTY720 restored PP2A activity by suppressing SE translocation (SET) and the cancerous inhibitor of PP2A (CIP2A), two endogenous PP2A inhibitors, independently of its antagonistic activity towards S1PRs [20]. PP2A activation significantly inhibits growth in CRC cell lines in a dose-dependent manner [26]. Another study showed that autophagy is involved in FTY720-mediated cytotoxicity in CRC cell lines, since treatment with 3-methyladenine (3-MA), an inhibitor of autophagy, heightened FTY720 cytotoxicity. This suggests that autophagy has a protective role against the drug’s own cytotoxic effect [27].

Other less recognized effects of FTY720 in CRC cells include downregulation of cyclin D1 and the phosphoinositide-3-kinase/Akt/mammalian target of rapamycin (PI3K/Akt/mTOR) signaling pathway, dephosphorylation of extracellular signal-regulated kinase (ERK), and inhibition of SphK [28]. The large number of downstream molecular targets of FTY720 reflect its potential as an anticancer drug, which could be combined with other therapies to overcome drug resistance and achieve better outcomes in patients.

FTY720 is known to have minimal cytotoxic effects on normal colonic cells [29], and could, therefore, possibly have less adverse effects in cancer patients. Selectively targeting cancer cells with low PP2A activity is attractive, since PP2A activators have minimal effects on cells with normal PP2A activity [19]. Discovering biomarkers of PP2A activity and, hence, sensitivity to PP2A activators such as FTY720 will facilitate the stratification of patients in order to select those more likely to respond to the treatment.

Aurora kinase A (AURKA) mRNA expression was identified as a biomarker of PP2A-dependent growth factor activation, and was used to classify breast cancer cases into a novel therapeutic class of tumors with predicted susceptibility to the restoration of PP2A activity using FTY720 (unpublished data). It would be of interest to evaluate whether AURKA expression might offer a similar opportunity to predict novel therapeutic subtypes in CRC.

The main aim of this study was to assess AURKA expression in CRC cell lines as a predictive biomarker of sensitivity to FTY720, and to group CRC cell lines into sensitive and non-sensitive groups following exposure to FTY720. 

## 2. Materials and Methods

### 2.1. Cell Cultures

The human CRC cell lines LS411N (CRL-2159), T84 (CCL-248), HCT116 (CCL-247), SW48 (CCL-231), LOVO (CCL-229), NCI H716 (CCL-251), COLO 320DM (CCL-220), DLD-1 (CCL-221), SW480 (CCL-228), and HT-29 (HTB-38), acquired from the American Type Culture Collection (ATCC, Manassas, VA, USA), were cultured in Dulbecco’s Modified Eagle Medium (DMEM)-high glucose (Sigma-Aldrich, St. Louis, MO, USA) with 10% foetal bovine serum (FBS) (Sigma-Aldrich). They were grown at 37 °C in a 5% CO_2_ atmosphere. The media were supplemented with penicillin G (100 U/mL) and streptomycin (0.1 mg/mL). 

### 2.2. RNA Extraction

RNA extraction was carried out using the RNeasy^®^ Mini Kit (Qiagen, Valencia, CA, USA), and quantified by 260/280 nm using UV-spectrophotometry (Nanodrop^®^, Thermo Fisher Scientific, Waltham, MA, USA). The quality of the RNA was determined by the RNA integrity number (RIN) using the RNA6000 Nano protocol on the Agilent 2100 Bioanalyzer system (Agilent, Santa Clara, CA, USA). 

### 2.3. Gene Expression Assay

The mRNA was isolated from the mentioned CRC cell lines and analyzed using the QuantiGene^®^ Plex gene expression assay (Thermo Fisher Scientific) [30]. Gene expression of the PP2A activity biomarker (PAB) AURKA together with the expression of three reference genes (TBP, HPRTI1, and PPIB) were measured. The mean of the blank was subtracted from the gene expression data extracted from the QuantiGene^®^ Plex gene expression assay. The limit of detection (LOD), defined as the sample that yields a signal higher than three times the standard deviation of the mean of the blank (3σ, 99% confidence level), was calculated. The gene expression data of AURKA were divided by the geomean of the gene expression data of the reference genes. 

Based on the Quantigene^®^ Plex gene expression assay results, three cell lines predicted to be sensitive to FTY720 (HT-29, DLD-1, and SW480) and three cell lines predicted to be non-sensitive to FTY720 (T84, LS411N, and HCT116) were cultured with various concentrations of FTY720.

### 2.4. FTY720 Preparation

The FTY720 was purchased from Sigma-Aldrich, and dissolved in 1% dimethyl sulfoxide (DMSO, Sigma-Aldrich) to a primary concentration of 58 mM. Working solutions of FTY720 and DMSO (vehicle control) were prepared fresh in the culture medium.

### 2.5. Cell Viability Assays

Cell viability assays were carried out on six CRC cell lines, of which three were predicted to be sensitive to FTY720, and three were predicted to be non-sensitive to the PP2A activator. The CRC cell lines were seeded in a 96-well plate, and treated with 0.05 µM, 0.1 µM, 0.5 µM, 1 µM, 2.5 µM, 5 µM, 10 µM, and 25 µM FTY720, and cell survival rate was evaluated using a standard 3-(4,5-dimethylthiazol2-yl)-2,5-diphenyltetrazolium bromide (MTT) assay (Sigma-Aldrich). After 24 h, the medium was aspirated, and fresh medium containing MTT (5 mg/mL) was added to each well. The cells were incubated at 37 °C for 4 h, after which, the plates were centrifuged at 14,000× *g* for 10 min. The supernatant was decanted, and 100 μL of DMSO was added to each plate. The plate was then placed on a thermomixer (Eppendorf, Hamburg, Germany) for 2 min at 300 rpm to ensure that all crystals dissolved, and then transferred to a spectrophotometer (Mithras LB 940 Multimode Microplate Reader, Berthold Technologies, Bad Wildbad, Germany), and the absorbance at 570 nm was measured. The same procedure was repeated after 48 h. This procedure was repeated twice. 

### 2.6. RNA Sequencing

RNA samples were submitted for library generation and sequencing to Beijing Genomics Institute (BGI). In brief, enrichment of poly(A) mRNA was performed using poly(T) oligo attached to magnetic beads, followed by fragmentation. First strand cDNA synthesis was carried out using random hexamer N6 primers and reverse transcriptase, followed by adaptor ligation to cDNA fragments. Following PCR amplification and purification, single-stranded DNA circles were generated in a final library. DNA nanoballs (DNBs) were subsequently generated by rolling circle replication, which underwent paired end sequencing (100 bp) on the DNBseq^®^ platform (BGI, Shenzhen, China).

The raw image data produced by the sequencer was converted by BGI into sequences using base calling software. Data filtering was then carried out to obtain ‘clean reads’ facilitating correct alignments in downstream analysis. This included the identification and removal of adaptor sequences, as well as the removal of low quality reads, and reads with lengths smaller than the set threshold. The average quality of reads of 20 bases was calculated from the 3′-end until the average quality was larger than nine. The bases that negatively impacted the quality in the 3′-end were removed. Assessment sequencing was conducted where the distributions of the read lengths were plotted and sequenced. 

The gene expression level was calculated using reads per kilobase per million mapped reads (RPKM), which is a normalisation technique used to eliminate any variation caused by differences in sample quality and coverage. The sequencing depth was normalised by dividing the total read counts in a sample by a scaling factor of one million, resulting in reads per million mapped reads (RPM). The gene length was also normalised by dividing the RPM values by the length of the gene in kilobases.

The resultant RPKM values were used to compare the difference in gene expression among samples. When multiple transcripts for a gene were present, the longest transcript was used to calculate its expression level and coverage. The analysis of differentially expressed genes (DEGs) was conducted based on the analysis method of the Poisson distribution. This included the screening of genes that were differentially expressed among the samples analyzed. The Gene ontology (GO) and Kyoto Encyclopedia of Genes and Genomes (KEGG) pathway enrichment analysis was then conducted.

### 2.7. Statistical Analysis

All statistical analysis was carried out using Microsoft Excel or the SPSS statistical package v23 (IBM SPSS, Chicago, IL, USA). Given that the gene expression data extracted from the QuantiGene^®^ Plex gene expression assay was non-parametric, the Mann–Whitney U test was used to test the hypothesis that AURKA mRNA is significantly overexpressed in CRC cell lines that are sensitive to the PP2A activator FTY720, compared to CRC cell lines that are non-sensitive.

## 3. Results

### 3.1. AURKA Is Differentially Expressed in CRC Cell Lines

AURKA expression was normalized against the geometric mean of the three reference genes TBP, HPRT1, and PPIB, and was used to predict PP2A activity. As shown in Figure 1, the CRC cell lines LS411N, T84, HCT116, SW48, and LOVO had a lower relative AURKA expression than the median, and were predicted to be non-sensitive to FTY720. On the other hand, NCI H716, COLO320DM, DLD-1, SW480, and HT-29 showed a higher relative AURKA expression than the median, and were predicted to be sensitive to FTY720. The top three relative AURKA expressors, DLD-1, SW480, and HT-29, were hence selected for FTY720 assays, together with LS411N, T84, and HCT116, which had the lowest relative AURKA expression levels.

### 3.2. AURKA Expression Does Not Predict Sensitivity to FTY720

Cell viability assays were carried out to test whether AURKA expression predicted FTY720 sensitivity. The dose-dependent effect of FTY720 on CRC cell lines provided information on their sensitivity to PP2A activation. Sensitivity to FTY720 was defined as reaching IC_50_ at a dose lower than 5 µM.

The results of the cell viability assays after 24 and 48 h of exposure to FTY720 are illustrated as line graphs in Figure 2, where the percentage cell viability is expressed as a percentage of the vehicle control across the concentrations of FTY720. LS411N, T84, HCT116, and SW480 were sensitive to FTY720, whereas DLD-1 and HT-29 were non-sensitive. The percentage of viable cells of the vehicle control of each cell line was never under 85% of the untreated cells’ viability, thus ruling out any interference by the vehicle.

The Mann–Whitney U test was used to test the hypothesis that AURKA mRNA is significantly overexpressed in CRC cell lines that are sensitive to FTY720, compared to CRC cell lines that are non-sensitive. The difference in AURKA expression between the sensitive and non-sensitive groups was found to be non-significant using the Mann–Whitney U test (*Z* = −1.389, *p* = 0.165), leading us to accept the null hypothesis that AURKA mRNA expression does not predict sensitivity to FTY720. Based on the results of the cell viability assays, two sensitive and two non-sensitive steady state CRC cell lines were sent for RNA sequencing. 

### 3.3. Differential Expression of Sphinogolipid Pathway Genes Is Enriched in FTY-Sensitive CRC Cell Lines

The gene expression of two sensitive and two non-sensitive CRC cell lines for FTY720 were investigated by RNA sequencing. We decided to select LS411N (CMS1) and HCT116 (CMS4) as representative of the sensitive group, since both were derived from primary adenocarcinomas of the colon. T84 was not selected because it was derived from lung metastatic tissue. The DEGs were obtained by comparing the sensitive CRC cell lines against the two non-sensitive ones, which were DLD-1 (CMS1) and HT-29 (CMS3). We found 253 genes that were significantly altered in expression, including 195 upregulated and 58 downregulated genes (with a fold-change ≥ 1 or ≤ 0.5, respectively; adjusted *p*-value (*p*_adj_) < 0.05). For an overview of the DEGs, volcano plots were drawn where the fold-change of gene expression was plotted on the *x*-axis versus the significance of difference in gene expression between pools on the *y*-axis (Figure 3). The top five upregulated genes were ELOVL5, CHST15, HS6ST2, ARHGEF10, and B4GALNT4, whereas the top five downregulated genes were AFAP1-AS1, TMEM178B, AMIGO2, SCEL, and FFAM174B.

The 253 DEGs were clustered based on the GO and KEGG analysis. The top ten clusters in biological processes, cellular components, and molecular functions were obtained. The GO functional enrichment analysis demonstrated that DEGs were significantly enriched in the functional categories associated with proximal and distal pattern formation, protein serine/threonine kinase activity, and mitogen-activated protein kinase (MAPK) activity. The KEGG database was used to further identify critical signal regulation pathways. The top significant KEGG pathways were identified, and are shown in Figure 4 and Table 1. Most of the genes were involved either in the MAPK signaling pathway or in the sphingolipid signaling pathway. In the latter, the most significantly upregulated genes were CERS4, PPP2R2C, GNAZ, PRKCG, BCL2, MAPK12, and MAPK11.

## 4. Discussion

The results of this study suggest that CRC cell lines have a differential response to the PP2A activator FTY720, and that a subset of CRC cellular models have deregulated pathways that promote sensitivity to PP2A activation. Although previous studies in breast cancer cellular models showed that sensitivity to low-dose FTY720 could be predicted by a high mRNA expression of AURKA (unpublished data), this study shows that sensitivity to FTY720 in CRC cell lines cannot be predicted using this model. In addition, breast cancer cell lines are responsive to lower doses of FTY720 [31], suggesting a different mechanism of action of the drug.

The sensitive cell lines, which reached IC_50_ at doses of FTY720 less than 5 µM, were HCT116, T84, LS411N, and SW480. Since DLD-1 and HT-29 reached IC_50_ at doses greater than 5 µM, we classified them as non-sensitive to FTY720. Although in other studies, SW480, HT-29, and DLD-1 cell lines were reported to be sensitive to 10 µM of FTY720 [20], FTY720 is considered to be cytotoxic at these concentrations [19]. 

The DEGs obtained by comparing the sensitive and non-sensitive groups (as classified in this study) suggest that the main driver of sensitivity to FTY720 in CRC cells is associated with the sphingolipid pathway, represented by the CERS4, PPP2R2C, and BCL2 genes. Ceramide is known to be a tumor suppressor, promoting PP2A phosphatase activity [32], and leading to dephosphorylation of the anti-apoptotic protein Bcl-2 [33]. Studies show that the expression of CERS4 and other ceramide synthase genes is significantly deregulated in CRC [34]. The transcript levels of the sphingolipid pathway effectors are significantly upregulated in the FTY720-sensitive cell line group, suggesting the sensitisation of a subset of CRC cell models to PP2A-dependent, FTY720-induced cell death. Upon addition of FTY720, binding of phosphorylated-FTY720 to S1PR1 leads to the downregulation of survival-promoting signals [35,36]. Overexpression of CERS4 and its product, ceramide synthase, leads to increased ceramide synthesis, which consequently promotes apoptosis [37]. It is imperative to investigate the expression of ceramide and the activation of the PP2A complex at protein level in future studies to confirm this proposed mechanism of action in CRC cell lines. 

Sphingolipids are a family of molecules enriched in lipid rafts whose metabolites are emerging as bioactive signaling molecules involved in the development of cancer. S1P plays a pivotal role in the early stages of colorectal carcinogenesis: it increases cell proliferation, inhibits apoptosis, and promotes oncogenic transformation. The two isoforms of SphK, SphK1, and SphK2, catalyse the conversion of the membrane phospholipid sphingosine to the bioactive lipid S1P, an oncogenic mediator which drives a number of vital processes in tumor cells, including cell growth, survival, migration, invasion, and angiogenesis [38]. SphK1 regulates tumor cell proliferation, apoptosis, and invasion in CRC by inducing epithelial-mesenchymal transition (EMT) through the focal adhesion kinase/Akt/matrix metalloproteinase (FAK/Akt/MMP) axis, and by suppressing p38 and stress-activated protein kinases/Jun amino-terminal kinases (SAPK/JNK) signaling [39,40].

Expression levels of SphK1 are significantly higher in CRC cells and tissues compared to normal colonic mucosa, and are associated with more advanced tumor stages and a poorer prognosis in CRC patients [23,41,42,43,44]. SphK1 overexpression was found to occur in particular CRC cell lines resistant to epidermal growth factor receptor (EGFR)-targeted therapy, such as cetuximab, with the exception of HCT116 cells. FTY720 was effective in restoring the ability of cetuximab to induce apoptosis and suppress EGFR-dependent signal transduction in resistant CRC cell lines and in vivo models [45]. The mechanism could depend on cross-talks known to exist between EGFR-dependent pathways and intracellular S1P signaling [46,47,48]. Of interest, phosphorylated-FTY720 was shown to inhibit colitis-associated CRC growth and proliferation by suppressing SphK1 and S1P1 receptor expression [35].

Expression of SphK2 and the S1P transporter sphingolipid transporter 2 (SPNS2) are also upregulated in CRC specimens [49,50]. SphK2 appears to act in synergy with protein kinase D (PKD) to confer resistance to chemotherapies. Knocking down PKD using specific siRNA or exposure to the ERK inhibitor U0126 increased the sensitivity of HCT116 cells to the physiological anti-CRC agent sodium butyrate (NaBt) [51,52]. SPNS2 promotes proliferation, migration, and invasion, and inhibits apoptosis by regulating the S1P/S1PR1/3 axis, and activating the PI3K/Akt/mTOR and MAPK pathways in SW480 and HCT116 CRC cells [49]. In addition, SPL and S1P phosphatase, which play an important role as gatekeepers of carcinogenesis, are highly expressed in enterocytes, but are downregulated in CRC tissues, suggesting that CRC cells manifest a block in S1P catabolism [53,54].

Downregulation of SphK1 expression sensitized RKO cells to cisplatin (DDP) in a concentration and time-dependent manner [55]. Similarly, exposure to the SphK1-selective inhibitor-compound 5c attenuated the PI3K/Akt/mTOR cell survival signaling pathway in CRC cells, and enhanced their sensitivity to 5-FU [44]. PF-543, another SphK1 inhibitor, exerted potent anti-proliferative and cytotoxic effects in HCT116, HT-29, and DLD-1 cells, and significantly suppressed growth in HCT116 xenografts [56]. The SphK2 inhibitor ABC294640 induced apoptosis in transformed and primary CRC cells, and increased sensitivity to 5-FU and cisplatin [57]. Combination regimens based on a SphK1 or SphK2 inhibitor and FTY720 could, therefore, be a useful strategy for treating therapeutic-resistant CRC.

The pathogenesis of CRC may, therefore, be mediated by ceramide and members of the SphK/S1P pathway, which could represent selective targets for chemoprophylaxis. 

## 5. Conclusions

The results of this study show that the expression levels of members of the sphingolipid pathway are associated with sensitivity to FTY720 in CRC cell lines, and show potential for use as predictive biomarkers. Further studies on larger panels of CRC cells are required to validate these results. There appears to be a significant difference in the mechanisms associated with low PP2A activity in breast cancer as compared to CRC cell lines. We previously showed that breast cancer cell lines are sensitive to doses of FTY720 between 0.05 µM and 0.1 µM [31], and that sensitivity can be predicted by overexpression of AURKA (unpublished data). Although AURKA expression failed to predict the sensitivity of CRC cell lines to the PP2A activator FTY720, it is evident that specific cell lines are sensitive to 5 µM FTY720, potentially due to the differential expression of genes involved in the sphingolipid pathway. The validity of predictive biomarkers should be assessed in the context of the specific disease, and, hence, the promotion of personalized medicine has a central role in improving therapeutic response [58,59]. This study clearly highlights the importance of stratifying patients on the basis of molecular markers to guide treatment decisions, and suggests that more studies are required to address the validation of known markers in different patient groups.

## Figures and Tables

**Figure 1 jcm-10-04999-f001:**
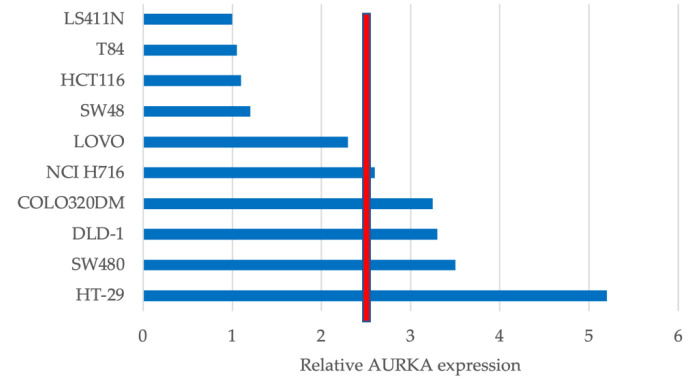
Relative Aurora kinase A (AURKA) expression in different colorectal cancer (CRC) cell lines. The median is represented by the red line. Based on AURKA expression, five cell lines, LS411N, T84, HCT116, SW48, and LOVO, were predicted to be non-sensitive to FTY720. NCI H716, COLO320DM, DLD-1, SW480, and HT-29 were predicted to be sensitive to FTY720.

**Figure 2 jcm-10-04999-f002:**
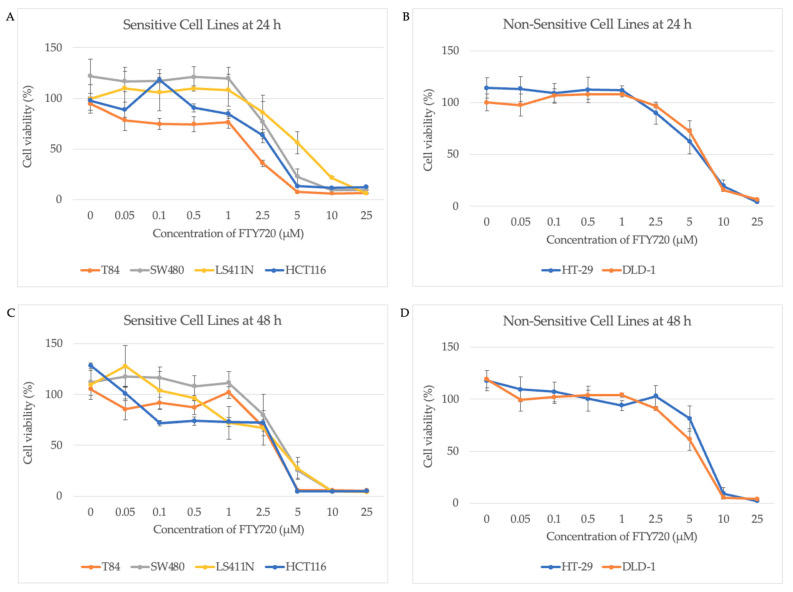
Line graphs of the cell viability assays. The cell viability of the cell lines at each dose of FTY720 is expressed as a percentage of the vehicle control. Data at 0 μM represent incubation of the cell lines in the medium without the vehicle (dimethyl sulfoxide (DMSO)). Error bars represent the standard deviation from three experimental replicates, each with three analytical repeats. (**A**) Sensitive colorectal cancer (CRC) cell lines at 24 h. (**B**) Non-sensitive CRC cell lines at 24 h. (**C**) Sensitive CRC cell lines at 48 h. (**D**) Non-sensitive CRC cell lines at 48 h.

**Figure 3 jcm-10-04999-f003:**
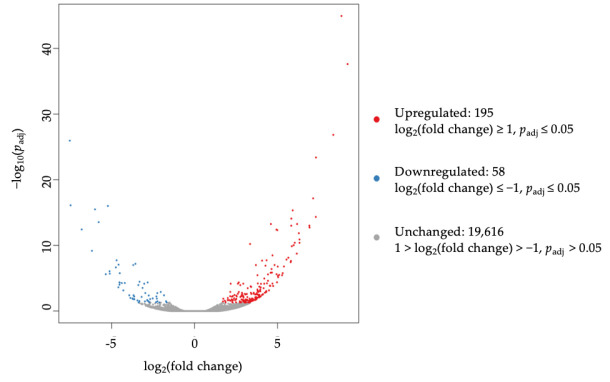
Volcano plot of differentially expressed genes (DEGs) between sensitive and non-sensitive colorectal cancer (CRC) cell lines. The *x*-axis represents the distribution of the log_2_ of the fold change, and the *y*-axis represents the −log_10_ of the adjusted *p*-value (*p*_adj_). Upregulated genes are shown in red, and downregulated genes are shown in blue.

**Figure 4 jcm-10-04999-f004:**
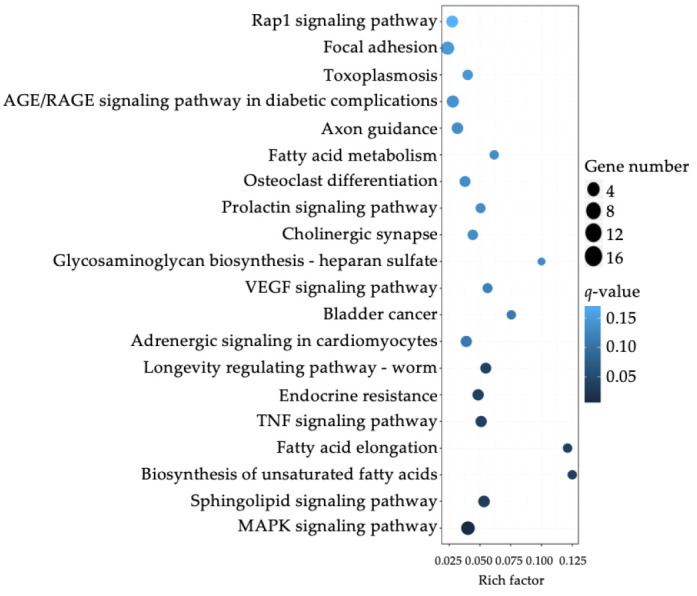
The 20 most enriched Kyoto Encyclopedia of Genes and Genomes (KEGG) pathways. Rich factor refers to the ratio of the number of enriched differentially expressed genes (DEGs) in the KEGG category to the total genes in that category. The larger the Rich factor, the greater the degree of enrichment. (AGE/RAGE = advanced glycation end-products/receptor for advanced glycation end-products. VEGF = vascular endothelial growth factor. TNF = tumor necrosis factor. MAPK = mitogen-activated protein kinase.)

**Table 1 jcm-10-04999-t001:** The top overrepresented pathways identified through impact analysis of differentially expressed genes (DEGs) when comparing the sensitive and non-sensitive colorectal cancer (CRC) cell lines.

Index	Pathway	*p*-Value	Adjusted *p*-Value (*p*_adj_)	Odds Ratio	Combined Score
1	Biosynthesis of unsaturated fatty acids	0.0000200	0.00286	18.08	195.57
2	Bladder cancer	0.00175	0.0502	8.56	54.31
3	Fatty acid elongation	0.00468	0.0912	9.86	52.91
4	VEGF signaling pathway	0.000893	0.0348	7.35	51.62
5	Sphingolipid signaling pathway	0.000138	0.0107	5.78	51.36
6	AGE/RAGE signaling pathway in diabetic complications	0.000281	0.0164	6.01	49.17
7	MAPK signaling pathway	0.0000245	0.00287	4.07	43.24
8	Nitrogen metabolism	0.0191	0.193	10.48	41.47
9	Prolactin signaling pathway	0.00193	0.0502	6.10	38.16
10	Growth hormone synthesis, secretion and action	0.000803	0.0348	4.99	35.55

VEGF = vascular endothelial growth factor. AGE/RAGE = advanced glycation end-products/receptor for advanced glycation end-products. MAPK = mitogen-activated protein kinase.

## Data Availability

This manuscript did not report any external data.

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
