# Peer review of "Differential Expression of the Sphingolipid Pathway Is Associated with Sensitivity to the PP2A Activator FTY720 in Colorectal Cancer Cell Lines"

_jcm, 2021, doi:10.3390/jcm10214999_

Round 1

Reviewer 1 Report

The Authors presented a very interesting paper describing a potent marker useful in cancer diagnosis. Nonetheless, the manuscript requires some corrections before its publishing. 

  1. The PP2A is doubled in abstract
  2. CRC is abbreviated in the abstract first. However, this should be also provided in the introduction.
  3. Please provide the company for each cell line used.
  4. Please provide in the methodology section the final concentration of DMSO used as a solvent, as this compound is known for its toxicity (cytotoxicity) above specific concentrations.
  5. Do the Authors posses any results regarding pH impact on cell viability and  FTY720. In fact, the Authors provided results indicated time (24 vs 48 h incubation) and dose-dependency. However, what was the pH of the studied cell lines exposure to FTY720? is it possible that higher concentrations of FTY720 resulted in some, even minor, changes in pH of the entire environment for the specific cell line?
  6. On what basis did the Authors choose for further study HCT and LS lines as those representative as sensitive CRC? This should be stated in the text.
  7. Since the Authors indicated that several used cancer cell lines are sensitive or not to FTY720, in my opinion, more detail information/characteristics should be introduced of these cells (similarities etc.)
  8. English language requires some corrections as punctuation and style errors are observed. In line to this, please standardize some words in the entire text, e.g. anti-cancer vs. anticancer

Reviewer 2 Report

The manuscript provided by the authors is a preliminary, but still interesting, study focused on the identification of molecular pathways that might predict the sensitivity of CRC cells to FMY720, a molecule whose effects on cell proliferation are well known and thus may represent a useful therapeutic approach for CRC. Although the findings are very preliminary and the types of performed experiments quite basic, they have been carried out on a quite large panel of CRC cell lines, which make the obtained results sufficiently reliable. Additionally, the topic in itself is interesting and the findings possess the potential to be applied in the clinical settings. Overall, I lean towards publication. However, the manuscript suffers of some flaws that must be corrected before the publication.

The first problem is the title, which is too ambitious not reflects the actual content of the manuscript. I understand the willing of using a tempting title, but it should at the same time gives to the reader a clear picture of what the work talks about, and this is not the case. In their manuscript, the authors do not identify a pathway able to predict FTY720 response in CRC cell line, they just identify a pathway associated with the response, and this is a very important difference. Base on their findings, the authors cannot state that this pathway is predictive, since they do not assess the predictivity power of the pathway itself. In order to do this, the authors had to test an additional panel of cell lines (primary and/or established), check for the expression of DEGs in these cells, divide them in hypothetical sensitive and not-sensitive cells and then assess if such prediction correlated with the actual sensitivity to FTY720. I guess it was beyond the scope of the work, and the authors appear to be quite aware of the limits of their study: indeed, abstract and conclusions sound far less ambitious compared to the title. Thus, the title should be changed clearly indicating that the identified pathway is associated with drug sensitivity.

The second problem concerns the discussion. Considering that the findings are mainly descriptive and no molecular mechanisms were investigated in order to explain the association, these had to be discussed in the discussion. Instead, the authors provide a very “descriptive” discussion that sounds more like a “list of evidences” than a critical evaluation of the obtained results. Indeed, the authors provide several evidences that FTY720 can affect the sphingolipid pathway by different mechanisms in different experimental systems, which represents the biological rationale of the reported association, but they do not discuss whether such mechanisms and which of them may occur in CRC cells. Reading the discussion, it sounds like all the mentioned mechanisms are equally likely to occur in CRC, and this approach is too vague and generic. The authors should be more critical in discussing the potential mechanisms underlying their findings, reporting the evidences supporting the hypothesis that a certain mechanism might occur in CRC or not.

Last but not least, the quality of the text should be improved. I noticed many refuses, typos and past tenses inconsistencies. I reported the mistakes I found below, but I strongly recommend the authors to carefully check the manuscript:

Line 7: please correct the refuse

Line 9: it should be “we have previously shown”

Line 10: please remove the comma

Line 17: it should be “non-sensitive CRC cell lines”

Line 21: please remove “the”

Line 22: it should be “potentially because of the…”

Line 28: please correct the refuse and mention the entire name

Lines 33-35: this sentence is not completely clear and should be rewritten

Line 55: the last point needs a reference

Line 59: please correct the refuse

Line 60: it should be “in CRC patients”

Line 62: please remove the comma

Line 64: please remove the comma

Line 66: please remove the comma

Lines 74-77: please provide the full names of SET and CIP2A as well as more details about their function(s)

Line 78: it should be “autophagy has been shown to be involved”

Lines 79-81: this sentence has been copied and pasted from the abstract of the related reference and should be thus rewritten in an original way

Line 93: it should be “in order to select those more likely respond to the treatment”

Lines 94-98: it is intuitable that AURKA is the gene for the coding Aurora A protein, but it is not made explicit well in the text. The period should be rewritten highlighting this point

Line 97: Ref 14 does not match with the content of the related sentence. Ref 14 refers to a review focused on CRC while the sentence refers to breast cancer. The authors should add the correct original reference

Line 99: it should be “to predict the sensitivity”

Line 124: please report the full name of HGs, not just the acronym

Line 130: please remove the comma

Line 139: the specific supplementary table is not indicated and the file is not provided

Line 200: it should be “predicted”

Line 201: it should be “provided”

Lines 204-210: MTT results at 24h and with/without vehicle should be shown in Figure 2 or as supplementary figure; otherwise, it is impossible to assess what the authors state. Also, it is not clear the criterion used by the authors to choose the cell lines on which MTT assay was carried out. This should be explained in the text

Figure 2/lines 212-217: the letters identifying the panels are reversed in the caption, which should be amended. It should be “is expressed as percentage compared to the control vehicle”.

Line 218: it should be “was”

Lines 223-225: the selected cell lines should be indicated here. Also, the authors should better explain the criterion behind the choice of the sensitive cell lines, since it is not very clear to infer from Figure 2. Indeed, the dose responses of the two cell lines are quite dissimilar and, for HCT116, not very consistent with the used concentrations (someone would expect a lower viability at 1 uM compared to 0.05 uM). Please provide more information about this point

Line 227: “the basal expression” is not very appropriate, since it suggests a comparison with a “stimulated” expression. It should be better to simply say “gene expression”

Figure 3: please increase the legend font. Also, it should be “p-values”

Figure 4: please increase the legend font.

Lines 265-267: these results should be shown in the Results section, not just discussed

Lines 277-278: this sentence is not consistent with what has been previously stated. The point is not the factors regulating Aurora A expression in CRC cell lines, but those, aside from Aurora A, affecting FTY720 sensitivity

Line 280: it should be “were”

Lines 282-284: this sentence is not supported by the subsequent discussion and references. Starting from the line 317, the authors report several evidences indicating that FTY720 may be involved in sphingolipid pathway. Thus, the authors should work on the inner consistency of the discussion

Line 285: it should be “enriched”

Line 303: please use the acronym only

Line 315: it is unclear what the authors mean with “chemoprevention”. This point should be better explained

Lines 315-316: this statement needs a reference. If it is just a hypothesis, it should be clearly stated

Line 327: please explain what FTY720-P is

Line 331: it should be “mechanisms”, not “effects”

Lines 330-331: Ref 47 refers to prostate cancer and S1P-indipendent mechanisms only. More references should be provided in order to support the statement

Lines 331-333: the provided reference (28) does not refer to neuroblastoma. Please provide a correct reference

Line 357: correct reference is the number 60. Please correct it

Line 361: “sphingolipid profile” is not correct in this context since the signalling pathway was investigated, not the sphingolipids within the cells

Line 365: it should be “we previously showed”

Line 369: it should be “potentially”

Lines 370-372: this sentence is not clear

Line 382: it should be “any external data”

Round 2

Reviewer 2 Report

I really appreciated the efforts made by the authors to address my questions. I think that the results section is overall fine and the discussion section has been markedly improved. However, some further improvements are required for discussion regard the following points: 

Page 12: the sentence "The high level of transcripts representing the sphingolipid pathway in the sensitive cell lines suggest that, upon exposure to FTY720, they become susceptible to PP2A-induced cell death" does not explain the correlation between sphingolipid pathway and sensitivity to FTY720. Did the authors want to suggest that the sensitivity might belinked to the sphingolipid pathway? This point should be better explained

Page 13: the part related to the capability of FTY720 to restore the sensitivity to cetuximab should be moved up, where the authors discuss the capapbility of sphingolipid pathway to blunt cetuximab anti-tumor effects

Pages 13-14: the parts related to SphK1 and SphK2 should be moved up where these enzymes are discussed for the first time, in order to immediately highlight the potential activity of FTY720 against them

Minor corrections:

Page 1: it should be "were shown to be sensitive to..."

Page 2: it should be "in treating CRC patients"

Page 3: before ref. 21, "inhibition is repeated twice, please correct the refuse

Page 7: it should be "the percentage of cell viability"

Page 7: it should be " was never under 85% of untreated cells' viability" 

Figure 3: please increase the legend font  (Fold LongChanges) of at least two points because it is very difficult to read

Figure 4: please increase the legend font  (Gene Number and Q-values) of at least two points because it is very difficult to read

Figure 4 caption: it should be "and the number of genes that has been..."

Page 11: it should be "studies in breast cancer cellular models showed that sensitivity to low-dose could be predicted by a high mRNA expression..."

Page 12: The sentence "In addition, breast cancer cell lines are responsive to lower doses of FTY720, suggesting a different mechanism of action of the drug." needs a reference

Page 12: it should be "that the sensitivity to FTY720 in CRC cells is associated with"

Page 15: it should be "and hence the promotion of
personalized medicine has a central role to improve therapeutic response"

Since the major part of the points have been correctly addressed, I would kindly ask the authors to upload the new version of the manuscript without tracked changes, In this way, it would be easier to read the discussion and to evaluate the overall improvement

Author Response

Page 12: the sentence "The high level of transcripts representing the sphingolipid pathway in the sensitive cell lines suggest that, upon exposure to FTY720, they become susceptible to PP2A-induced cell death" does not explain the correlation between sphingolipid pathway and sensitivity to FTY720. Did the authors want to suggest that the sensitivity might belinked to the sphingolipid pathway? This point should be better explained

Explained the association as follows: The transcript levels of the sphingolipid pathway effectors is significantly upregulated in the FTY720-sensitive cell line group, suggesting the sensitisation of a subset of CRC cell models to PP2A-dependent, FTY720-induced cell death. Upon addition of FTY720, binding of phosphorylated-FTY720 to S1PR1 leads to the downregulation of survival-promoting signals [35,36]. Overexpression of CERS4 and its product ceramide synthase leads to increased ceramide synthesis, which consequently promotes apoptosis [37]. It is imperative to investigate the expression of ceramide and the activation of the PP2A complex at protein level in future studies to confirm this proposed mechanism of action in CRC cell lines.

Page 13: the part related to the capability of FTY720 to restore the sensitivity to cetuximab should be moved up, where the authors discuss the capapbility of sphingolipid pathway to blunt cetuximab anti-tumor effects

DONE.

Pages 13-14: the parts related to SphK1 and SphK2 should be moved up where these enzymes are discussed for the first time, in order to immediately highlight the potential activity of FTY720 against them

DONE

Minor corrections:

Page 1: it should be "were shown to be sensitive to..."

DONE

Page 2: it should be "in treating CRC patients"

DONE

Page 3: before ref. 21, "inhibition is repeated twice, please correct the refuse

DONE

Page 7: it should be "the percentage of cell viability"

DONE

Page 7: it should be " was never under 85% of untreated cells' viability" 

DONE

Figure 3: please increase the legend font  (Fold LongChanges) of at least two points because it is very difficult to read

DONE

Figure 4: please increase the legend font  (Gene Number and Q-values) of at least two points because it is very difficult to read

DONE

Figure 4 caption: it should be "and the number of genes that has been..."

Changed to ‘’and the number of genes in each pathway has been annotated’’.

Page 11: it should be "studies in breast cancer cellular models showed that sensitivity to low-dose could be predicted by a high mRNA expression..."

DONE

Page 12: The sentence "In addition, breast cancer cell lines are responsive to lower doses of FTY720, suggesting a different mechanism of action of the drug." needs a reference

DONE

Page 12: it should be "that the sensitivity to FTY720 in CRC cells is associated with"

DONE

Page 15: it should be "and hence the promotion of
personalized medicine has a central role to improve therapeutic response"

DONE
